# Generalized Expected Utility as a Universal Decision Rule – A Step Forward

**Hélène Fargier**[1, 2]  **Pierre Pomeret-Coquot**[1, 2]

[1]IRIT, Université de Toulouse, CNRS, Toulouse INP, UT3, UT2, Toulouse, France
[2]ANITI, B612, 3 rue Tarfaya, 31400 Toulouse

## Abstract

In order to capture a larger range of decision rules, this paper extends the seminal work of [Friedman and Halpern, 1995, Chu and Halpern, 2003, 2004] about Generalized Expected Utility. We introduce the notion of algebraic mass function (and of algebraic Möbius transform) and provide a new algebraic expression for expected utility based on such functions. This utility, that we call "XEU", generalizes Chu and Halpern's GEU to non-decomposable measures and allows for the representation of several rules that could not be captured up to this point, and noticeably, of the Choquet integral. A representation theorem is provided that shows that only a very weak condition is needed for a rule in order to be representable as an XEU.

## 1 INTRODUCTION

Many different rules for decision making under uncertainty have been introduced and characterized in the literature, following the seminal work of [Savage, 1954] and [Morgenstern and Von Neumann, 1944]. Halpern et al. [Friedman and Halpern, 1995, Chu and Halpern, 2003, 2004] have proposed in the early twenty first century a general algebraic expression that captures many of them, namely the rules that "respects utility" (a form of Savage third axiom) and "uniformity". This algebraic generalization enables to consider several decision rules at a glance, e.g. to compare one to another one, to identify which (axiomatic) characteristics of the decision behavior of the decision maker lead to a rule or to another one, and to express general algorithms the soundness of which can be proved once and for all from the properties of the decision rule [Perny et al., 2005, Pralet et al., 2007].

Chu and Halpern's algebraic decision rule, called Generalized Expected Utility (GEU), suits decision problems were the knowledge about the possible states of the world can be captured by a decomposable measure (that i.e. a measure that derives from a distribution). Nevertheless, it cannot capture a range of situations that do not obey this assumption, and noticeably many decision rules based on the Choquet Expected Utility [Choquet, 1954]–nor general Sugeno integrals [Sugeno, 1974].

In order to capture a larger scope of decision rules, and in particular Choquet integrals, the present paper extends the seminal work of Halpern et al. It introduces the notion of algebraic mass function (and of algebraic Möbius transform) and provides a new algebraic expression of expected utility. This expression, that we call "XEU", generalizes GEU to measures that are not necessarily decomposable. A representation theorem is provided that shows that only very weak conditions are needed for a rule in order to be representable as an XEU.

The paper is organized as follows. Section 2 details the background and motivations of this work. Section 3 then introduces the two main contributions of this paper: algebraic mass functions and the XEU decision rule. Section 4 is devoted to the exact representation (in terms of utility value) by XEU instances of several rules proposed in the literature. For the sake of readability, all the proofs are deferred to the Appendix.

## 2 BACKGROUND AND MOTIVATIONS

### 2.1 ALGEBRAIC CAPACITIES

The knowledge and/or belief of an agent about the actual state of the world is classically captured by a monotone set-function or "capacity measure". This function is not necessarily real-valued and may range over a partially ordered scale (for instance [Wilson, 1995]'s order of magnitude calculus). Hence the definition of algebraic capacities (called "Plausibility measures" in [Friedman and Halpern, 1995]) that map each event to the confidence degree the agent has

in the realization of that event. Formally:

**Definition 1.** *A plausibility domain is a set $W$ equipped with a reflexive, antisymmetric and transitive relation $\succeq^W$ ($\succeq^W$ is a partial order) admitting a minimum element $\perp_W$ and a maximal element $\top_W$ (i.e. such that $\forall w \in W$, $\top_W \succeq^W w \succeq^W \perp_W$).*

In the following, the subscripts and superscripts $W$ are omitted when clear from the context, e.g. we shall write $\top \succeq \perp$.

Given a finite set of states $\Omega$, an algebraic capacity is a set-function $\mu : 2^\Omega \to W$ which is monotone and pointed.[1]

**Definition 2** (Monotonicity). *A set-function $\mu : 2^\Omega \to W$ is monotone iff for any events $A \subseteq B \subseteq \Omega$, $\mu(A) \preceq^W \mu(B)$.*

**Definition 3** (Pointedness). *A set-function $\mu : 2^\Omega \to W$ is pointed iff $\mu(\emptyset) = \perp$ and $\mu(\Omega) = \top$.*

**Definition 4** (Algebraic capacity). *An algebraic capacity on $\Omega$ with co-domain $W$ is a set-function $\mu : 2^\Omega \to W$ that is both monotone and pointed.*

As shown by Friedman and Halpern [1995], several common measures are captured by algebraic capacities: probability measures, lower and upper probabilities [Walley and Fine, 1982], belief functions [Dempster, 1967, Shafer, 1976], fuzzy (Sugeno) measures [Sugeno, 1974], $\kappa$-rankings [Spohn, 1988], parameterized probability distributions [Goldszmidt and Pearl, 1992], among others.

**Example 1** (Algebraic capacities). *Let $\Omega = \{\omega_1, \omega_2, \omega_3\}$ and $\Pr$, $\mathrm{Bel}$ and $\Pi$ be the three set-functions listed in Table 1. Notice that $\Pr$, $\mathrm{Bel}$ and $\Pi$ are respectively a probability*

| $A$ | $\Pr(A)$ | $\mathrm{Bel}(A)$ | $\Pi(A)$ |
|---|---|---|---|
| $\emptyset$ | 0 | 0 | impossible |
| $\{\omega_1\}$ | 1/3 | 0 | totally possible |
| $\{\omega_2\}$ | 1/3 | 0 | somewhat possible |
| $\{\omega_3\}$ | 1/3 | 0 | impossible |
| $\{\omega_1, \omega_2\}$ | 2/3 | 1 | totally possible |
| $\{\omega_1, \omega_3\}$ | 2/3 | 0 | totally possible |
| $\{\omega_2, \omega_3\}$ | 2/3 | 0 | somewhat possible |
| $\{\omega_1, \omega_2, \omega_3\}$ | 1 | 1 | totally possible |

Table 1: Three capacities

*measure, a belief function and a possibility measure.*

- *$\Pr$ is an algebraic capacity, using $W = [0, 1]$, $\perp = 0$, $\top = 1$ and $\succeq^W = \geq$;*
- *$\mathrm{Bel}$ is an algebraic capacity, using the same settings;*

---

[1]Throughout the paper we assume that $\Omega$ is finite. This assumption echoes Chu and Halpern's assumption that the space of possible outcomes is finite.

- *$\Pi$ is also an algebraic capacity, using $W = \{impossible, \ somewhat \ possible, \ totally \ possible\}$, $\top = totally \ possible$, $\perp = impossible$ with impossible $\prec^W$ somewhat possible $\prec^W$ totally possible.*

Some algebraic capacities are decomposable, in the sense that the values they yield for an event can be retrieved from the values they yield for singletons only. Formally:

**Definition 5** (Decomposable capacity). *An algebraic capacity $\mu : 2^\Omega \to W$ is decomposable by an operator $\boxplus$ on $W$ (or "$\boxplus$-decomposable") iff for any disjoint events $A$ and $B$, it holds that $\mu(A \cup B) = \mu(A) \boxplus \mu(B)$.*

From this definition, it follows $\boxplus$ is necessarily commutative and admits $\perp$ as a neutral element. All the information contained in a decomposable capacity is captured by its restriction to singletons – when $\mu$ is decomposable then for any $A$, $\mu(A) = \boxplus_{\omega \in A} \mu(\{\omega\})$. This is the principle at work when computing a probability measure (resp. a possibility measure) on events from a probability distribution (resp. a possibility distribution). In other terms, probability measures (resp. possibility measures) are decomposable according to $\boxplus = +$ (resp. $\boxplus = \max$).

**Example 2** (Decomposable capacities). *Two of the capacities of Example 1 are decomposable, namely $\Pr$ (decomposable by $+$) and $\Pi$ (decomposable by $\max$).*
*On the contrary, $\mathrm{Bel}$ is not decomposable. Suppose indeed that there exists $\boxplus$ such that $\mathrm{Bel}(\{\omega_1, \omega_2\}) = \mathrm{Bel}(\{\omega_1\}) \boxplus \mathrm{Bel}(\{\omega_2\}) = 0 \boxplus 0 = 1$ and also such that $\mathrm{Bel}(\{\omega_1, \omega_3\}) = \mathrm{Bel}(\{\omega_1\}) \boxplus \mathrm{Bel}(\{\omega_3\}) = 0 \boxplus 0 = 0$, thus $0 = 1$: the hypothesis of decomposition of $\mathrm{Bel}$ leads to a contradiction.*

## 2.2 GENERALIZED EXPECTED UTILITY

In accordance with Savage's model [Savage, 1951, 1954], an act (a potential decision) is represented by a function $\chi : \Omega \to \mathbb{X}$, mapping each state of the world to a consequence (or "outcome"): $\chi(\omega)$ is the outcome that will be obtained if the actual state of the world is $\omega$.

A *decision situation* describes the "objective" components of a decision problem: it summarizes the possible choices a decision maker (DM) can make.

**Definition 6** (Decision situation). *A decision situation is a tuple $\mathcal{A} = (\Omega, \mathbb{X}, \mathbb{A})$ where:*

- *$\Omega$ is a finite set of states of the world;*
- *$\mathbb{X}$ is a finite set of outcomes;*
- *$\mathbb{A}$ is the finite set of feasible acts $\chi : \Omega \to \mathbb{X}$.*

Any two DMs that are facing the same problem should consider the same decision situation (they choose among the same set of acts) but the two DMs may obviously differ on their respective knowledge and preferences.

A decision rule is a function that captures the preferences of the DM about the acts when the knowledge about the world is pervaded with uncertainty. We have seen that the knowledge of the DM can be captured algebraically by a capacity $\mu : 2^\Omega \mapsto W$. The preferences about the outcomes may be represented in the same way, by a utility function $u$ mapping each outcome to an domain of utility $U$ ($U$ can be partially ordered, e.g. in applications involving several, non commensurable, criteria).

So, a decision problem involves an objective part – the decision situation–and a subjective part–the preferences of the DM about the outcomes and the knowledge of the DM about the state of the world.

**Definition 7** (Decision problem). *A decision problem is a tuple $\mathcal{D} = (\mathcal{A}, U, W, u, \mu)$ where:*

- *$\mathcal{A} = (\Omega, \mathbb{X}, \mathbb{A})$ is a decision situation;*
- *$U$ is a set equipped by a reflexive relation $\succeq^U$;*
- *$W$ is a plausibility domain;*
- *$u : \mathbb{X} \to U$ is a utility function;*
- *$\mu : 2^\Omega \to W$ is an algebraic capacity.*

Given a set of acts, a decision rule leads to a preference order on acts, or equivalently to a function $\mathbf{v}$ mapping acts to an (abstract) ordered scale $\mathrm{Img}(\mathbf{v})$:

**Definition 8** (Decision rule). *A decision rule $\mathbf{v}$ on a set of decision problems $\mathrm{Dom}(\mathbf{v})$ is a function that maps any decision problem $\mathcal{D} \in \mathrm{Dom}(\mathbf{v})$ and any act $\chi$ of $\mathcal{D}$ to a value $\mathbf{v}_\mathcal{D}(\chi)$ in a set $\mathrm{Img}(\mathbf{v})$ equipped with a reflexive relation $\succeq^{\mathbf{v}}$.*

Notice that the decision rule does not necessarily use the utility function $u$ and the capacity $\mu$. But many meaningful rules are based on the aggregation of the confidence and utility levels by means of some operators, say $\otimes$ and $\oplus$, the first one combining utility levels to confidence levels, the second one aggregating these elementary expected utilities – hence the proposition by Chu and Halpern [2003] of the "Generalized Expected Utility" induced by a decision problem.

**Definition 9** (Expectation domain). *An expectation domain is a tuple $E = (U, W, V, \oplus, \otimes)$ where:*

- *$W$ is a plausibility domain;*
- *$U$ is a set ordered by a reflexive relation $\succeq^U$ (the utility domain);*
- *$V$ is a set ordered by a reflexive relation $\succeq^V$ (the valuation domain);*
- *$\oplus : V \times V \to V$ is an associative and commutative operator;*
- *$\otimes : W \times U \to V$ is an operator such that $\top \otimes x = x$.[2]*

---

[2] $\top$ is typically the confidence into the realization of the certain events (of $\Omega$). Condition $\top \otimes x = x$ ensures that the global utility of an act of constant utility $x$ is equal to $x$.

An expectation domain is compatible with a decision problem as soon as they share the same utility domain ($U$) and the same plausibility domain ($W$).

**Definition 10** (GEU). *Let $\mathcal{D} = (\mathcal{A}, U, W, u, \mu)$ be a decision problem and $E = (U, W, V, \oplus, \otimes)$ be an expectation domain compatible with $\mathcal{D}$. The GEU of an act $\chi$ of $\mathcal{D}$ according to $E$ is:*

$$\mathrm{GEU}_\mathcal{D}^E(\chi) = \bigoplus_{x \in u(\chi(\Omega))} \mu(\{\omega \mid u(\chi(\omega)) = x\}) \otimes x,$$

*where $u(\chi(\Omega)) = \{u(\chi(\omega)) \mid \omega \in \Omega\}$ is the set of utility degrees reached by $\chi$.*

Then $\chi$ is at least as good as $\chi'$ iff $\mathrm{GEU}_\mathcal{D}^E(\chi) \succeq^V \mathrm{GEU}_\mathcal{D}^E(\chi')$.

In [Chu and Halpern, 2004], the authors cope with the following issue: representing a particular decision rule as a GEU instance–in more details, the problem is to find the adequate expectation domain $E = (U, W, V, \oplus, \otimes)$ for the GEU to yield the same preference relation on acts than the decision rule when provided with the same decision problem, i.e. the same decision situation, the same utility function and the same capacity. Since $\mu$ and $u$ (and thus $U$ and $W$) are given as inputs, the problem is actually to find $V$, $\oplus$ and $\otimes$.

**Definition 11** (GEU representation [Chu and Halpern, 2004]). *Let $\mathbf{v}$ be a decision rule and $E = (U, W, V, \oplus, \otimes)$ an expectation domain. $E$ is a GEU representation of $\mathbf{v}$ iff for any decision problem $\mathcal{D} \in \mathrm{Dom}(\mathbf{v})$ and for all acts $\chi$ and $\chi'$ of $\mathcal{D}$:*

$$\mathbf{v}_\mathcal{D}(\chi) \succeq^{\mathbf{v}} \mathbf{v}_\mathcal{D}(\chi') \iff \mathrm{GEU}_\mathcal{D}^E(\chi) \succeq^V \mathrm{GEU}_\mathcal{D}^E(\chi').$$

The authors show that the rules that admit an GEU representation are those that are "uniform" and "respect utility":

**Definition 12** (Respect of utility). *A decision rule $\mathbf{v}$ respects utility iff for any $\mathcal{D} \in \mathrm{Dom}(\mathbf{v})$ and for any two constant acts $\chi_1, \chi_2$ of $\mathcal{D}$ ($\chi_1(\omega) = x_1$ and $\chi_2(\omega) = x_2 \ \forall \omega$):*

$$\mathbf{v}_\mathcal{D}(\chi_1) \succeq^V \mathbf{v}_\mathcal{D}(\chi_2) \iff u(x_1) \succeq^U u(x_2).$$

**Definition 13** (Uniformity). *A decision rule $\mathbf{v}$ is uniform iff for all $\mathcal{D} \in \mathrm{Dom}(\mathbf{v})$ and all acts $\chi_1, \chi_2, \chi_1', \chi_2'$ of $\mathcal{D}$ such that $\mu(\{\omega \mid u(\chi_i(\omega)) = x\}) = \mu(\{\omega \mid u(\chi_i'(\omega)) = x\})$ for each $x \in U$, it holds that:*

$$\mathbf{v}_\mathcal{D}(\chi_1) \succeq^V \mathbf{v}_\mathcal{D}(\chi_2) \iff \mathbf{v}_\mathcal{D}(\chi_1') \succeq^V \mathbf{v}_\mathcal{D}(\chi_2').$$

**Theorem 1** (Chu and Halpern [2004]). *A decision rule admits a GEU representation iff it is uniform and respects utility.*

This theorem outlines the generality of the framework. Expected utility is of course an instance of GEU, with $U = \mathbb{R}$,

$W = [0, 1]$, $V = \mathbb{R}$, $\oplus = +$ and $\otimes = \times$. It is also the case of several rules working on decomposable measures, e.g. the possibilistic utilities [Dubois and Prade, 1995] and [Wilson, 1995]'s order of magnitude calculus. Among the rules for decision making under total ignorance, the maximin rule is captured with $U = \mathbb{R}$, $W = \{0, 1\}$, $V = \mathbb{R}$, $\oplus = \min$ and $\otimes = \times$: here all the states of $\Omega$ are totally and equally possible, i.e. $\mu(A) = 1 \forall A \neq \emptyset$ and $\mu(\emptyset) = 0$.

The limitation of the formalism also is highlighted by this theorem: the decision rules which are not uniform cannot admit a GEU representation. Among these rules, let us cite the rules based on the Choquet integral or the Sugeno integral, but for particular cases.[3]

# 3 GENERALIZING EXPECTED UTILITY – A STEP FORWARD

Several frameworks that cannot be captured as an instance of the GEU model, e.g. the Choquet integral in its full generality, involve non decomposable capacities, i.e. capacities that cannot be summarized by a distribution on states. That is why the model we propose is based on the notion of mass function, that generalizes distributions by mapping elementary units of confidence to sets (and not only to singletons).

## 3.1 ALGEBRAIC MASS FUNCTIONS

As a matter of fact, let us refer to Dempster-Shafer theory of evidence [Dempster, 1967, Shafer, 1976] where the notion of mass function is seminal. In this theory, a measure of belief, denoted Bel, is derived from a mass function $m : 2^\Omega \to [0, 1]$ which maps atomic beliefs to events: the total belief of an event $A \subseteq \Omega$ is defined by $\mathrm{Bel}(A) = \sum_{B \subseteq A} m(B)$. Reciprocally, the mass function can be deduced from the Bel measure (and more generally from any real-valued capacity) thanks to the Möbius inverse transform [Shafer, 1976].

**Definition 14** (Möbius inverse transform). *The Möbius inverse $m : 2^\Omega \to \mathbb{R}$ of a real-valued capacity $\mu : 2^\Omega \to \mathbb{R}$ is defined by:*

$$m(A) = \sum_{B \subseteq A} (-1)^{|A \setminus B|} \times \mu(B).$$

Whatever the real-valued set function $\mu$ considered one can recover $\mu$'s values from its Möbius inverse transform $m$: it holds that for any event $A$: $\mu(A) = \sum_{B \subseteq A} m(B)$.

[3]Uniformity requires that any two acts which reach the same utility degrees with identical beliefs must be considered as equivalent. A consequence of this requirement is that the capacity must be decomposable - in other terms, GEU cannot deal with non decomposable ones. Hence its incapacity to capture most of the Sugeno and Choquet integrals.

Such a notion of mass function has also been proposed for qualitative approaches: from monotonicity, any capacity $\mu : 2^\Omega \to W$ can be encoded by a (non-unique) qualitative mass function $\gamma : 2^\Omega \to W$ such that $\forall A \subseteq \Omega, \mu(A) = \max_{B \subseteq A} \gamma(B)$. A qualitative mass function called the qualitative Möbius Inverse may be deduced from the original capacity [Mesiar, 1997, Grabisch, 2016]:

**Definition 15** (Qualitative Möbius inverse transform). *The qualitative Möbius inverse $\gamma : 2^\Omega \to W$ of a monotone capacity $\mu : 2^\Omega \to W$ is defined, for any $A \subseteq \Omega$, by:*

$$\gamma(A) = \begin{cases} \mu(A) & \text{if } \forall \omega \in A, \mu(A \setminus \{\omega\}) < \mu(A) \\ \bot & \text{otherwise.} \end{cases}$$

Let us now generalize the approach and propose the notion of *algebraic* mass function with regard to a given operator $\boxplus : W^* \times W^* \to W^*$, where $W^*$ is a superset of the plausibility domain considered.

**Definition 16** (Algebraic mass function). *Given a capacity $\mu$ on a domain $W$, a function $m : 2^\Omega \to W^*$ with $W \subseteq W^*$ and an operator $\boxplus : W^* \times W^* \to W^*$, a function $m$ is a $\boxplus$-based mass function of $\mu$ iff for any $A \subseteq \Omega$:*

$$\mu(A) = \boxplus_{B \subseteq A} m(B)$$

It often happens that the domains of the capacity and the mass function coincide (this is the case when considering the max-transform of a possibility measure or the Möbius inverse transform of a Bel measure) but not necessarily – for instance, the Möbius inverse transform of a measure of lower probability may involve negative masses.

Since $\boxplus$ iterates over subsets in an arbitrary order, $\boxplus$ must be associative and commutative. It is a commutative monoid iff it also admits a neutral element $0_\boxplus$ (an element $0_\boxplus$ such that $\forall w, w \boxplus 0_\boxplus = w$). Finally, since $\mu$ is an algebraic capacity, it holds that:

- for any $A \subseteq B$, $\boxplus_{C \subseteq A} m(C) \preceq^W \boxplus_{C \subseteq B} m(C)$ (from the condition of monotonicity);
- $m(\emptyset) = \bot$ and $\boxplus_{A \subseteq \Omega} m(A) = \top$ (from the pointedness condition).

These conditions suggest that $\bot$ might be a neutral element of $\boxplus$—this is generally the case (e.g. 0 for $[0, 1]$-capacities when decomposed by the classical, $+$-based, Möbius transform). But the condition is not necessary.

**Definition 17** (Focal element). *An event $B \subseteq \Omega$ is a focal element of $m : 2^\Omega \to W^*$ for $\boxplus$ iff $m(B) \neq 0_\boxplus$.*

When there isn't any neutral element $0_\boxplus$, all events are focal elements. The values of a capacity which admits a $\boxplus$-based mass function can be recovered by considering only the

focal elements of this mass function: when $m$ is a $\boxplus$-based mass function of $\mu$,

$$\mu(A) = \bigboxplus_{B \subseteq A, m(B) \neq 0_\boxplus} m(B).$$

In other terms, the existence of a neutral element $0_\boxplus$ enables a shorter encoding of the mass function (among the $2^{|\Omega|}$ subsets of $\Omega$, only those receiving a non-zero mass are recorded).

$\boxplus$-based distributions are mass functions the focal elements of which are singletons:

**Definition 18** (Distribution). *$m$ is a $\boxplus$-based distribution iff $\forall B \subseteq \Omega, m(B) \neq 0_\boxplus \implies |B| = 1$.*

Notice that the existence of a $\boxplus$-based distribution for $\mu$ supposes the existence of a neutral element.

As a matter of fact, a probability distribution is obviously the $+$-based mass function of the associated probability measure (and is a $+$-based distribution) and a possibility distribution is the $\max$-based mass function of the corresponding possibility measure (and is a $\max$-based distribution).

**Example 3.** *Consider a capacity $\mu$ on $\Omega = \{\omega_1, \omega_2, \omega_3\}$ described in Table 2. In this example the scale $W = \{0, 2, 4, 6\}$ is used, with $\perp = 0$ and $\top = 6$ and it is easy to check that $\mu$ is a possibility measure.*

| $A$ | $\mu(A)$ | $m_1(A)$ | $m_2(A)$ | $m_3(A)$ |
|---|---|---|---|---|
| $\emptyset$ | 0 | 0 | 0 | 0 |
| $\{\omega_1\}$ | 6 | 6 | 6 | 6 |
| $\{\omega_2\}$ | 4 | 4 | 4 | 4 |
| $\{\omega_3\}$ | 2 | 2 | 2 | 2 |
| $\{\omega_1, \omega_2\}$ | 6 | 0 | -4 | 1/4 |
| $\{\omega_1, \omega_3\}$ | 6 | 0 | -2 | 1/2 |
| $\{\omega_2, \omega_3\}$ | 4 | 0 | -2 | 1/2 |
| $\{\omega_1, \omega_2, \omega_3\}$ | 6 | 0 | 2 | 2 |

Table 2: A capacity $\mu$ and some of its mass functions.

*Firstly, like all capacities, $\mu$ is a $\max$-based mass function of itself. This follows from its monotonicity property. Secondly, let us observe that $m_1$ is another $\max$-based mass function of $\mu$. Furthermore, it is a distribution (since 0 is neutral for $\max$ on $W$ and only singletons are focal); it is indeed the possibility distribution obtained through the qualitative Möbius transform. Lastly, $m_2 : 2^\Omega \to \mathbb{N}$ and $m_3 : 2^\Omega \to \mathbb{Q}^+$ are two other mass functions with codomains $W^* \neq W$. Specifically, $m_2$ is the $+$-based mass function obtained through the Möbius transform, and $m_3$ is a $\boxplus$-based mass function, where $\boxplus$ is the pseudo-product defined as $x \boxplus 0 = 0 \boxplus x = x$ and $x \boxplus y = x \cdot y$ otherwise.*

Recovering a capacity from its mass function according to $\boxplus$ is an easy task: simply apply Definition 16. On the contrary, computing a $\boxplus$-based mass function from a capacity

is generally not an easy task. Depending on $\mu$ and $\boxplus$, there may be zero, one or several $\boxplus$-based mass functions of $\mu$. As to get such a $m$ from $\mu$, one shall develop the expression of $\mu(A)$ (Definition 16) over the proper subsets of $A$:

**Proposition 1.** *$m$ is a $\boxplus$-based mass function of $\mu$ iff:*

$$\mu(\emptyset) = m(\emptyset), \quad and$$
$$\mu(A) = m(A) \boxplus \bigboxplus_{B \subsetneq A} m(B) \qquad \forall A \neq \emptyset.$$

Given a capacity $\mu$ Proposition 1 provides a system of equations the unknowns of which are the masses. One may be tempted to inductively build $m$ from $\mu$ from this system. However, it is not certain whether this is possible (the equations may be inconsistent for some $\mu$ and $\boxplus$), nor that there is a unique way to do it. However, as long as $\boxplus$ has an inverse operation $\boxminus$ (such that $w \boxplus w' \boxminus w' = w$), the mass function always exists and is unique. It can be provided by a deterministic inductive algorithm:

**Definition 19** (Mass function based on a commutative group). *Let $\mu : 2^\Omega \to W$ be a capacity and $(W^*, \boxplus)$ be a commutative group such that $W \subseteq W^*$. Let $0_\boxplus$ denote the neutral element of $\boxplus$ and $\boxminus$ its inverse. The $\boxplus$-Möbius transform of $\mu$, denoted $m_\mu^\boxplus$, is the mass function recursively defined by:*

$$m_\mu^\boxplus(\emptyset) = \mu(\emptyset) = \perp$$
$$m_\mu^\boxplus(A) = \mu(A) \boxminus \bigboxplus_{B \subset A} m_\mu^\boxplus(B) \qquad \forall A \neq \emptyset.$$

**Theorem 2.** *When $(W^*, \boxplus)$ is a commutative group, $m_\mu^\boxplus$ is the unique $\boxplus$-mass function of $\mu$.*

The quantitative and qualitative Möbius inverses are of course special cases of algebraic mass functions (with $\boxplus = +$ and $\boxplus = \max$). Applying Theorem 2, we recover the result of [Shafer, 1976] about the unicity of the $+$-mass function for real-valued capacities. On the contrary, some capacities may admit several qualitative ($\max$-based) mass functions—among them, the qualitative Möbius inverse transform which is, as shown by [Grabisch, 1997], the one which has the fewest number of, and smallest, focal elements.

More generally, if $\boxplus$ hasn't any inverse, there may be zero or several $\boxplus$-based mass functions.

**Example 4.** *In Example 3 two $\max$-based mass functions are identified for $\mu$: $m_1$ and $\mu$ itself. This is not surprising, given that $\max$ has no inverse. In contrast, addition over $\mathbb{N}$ does have an inverse; therefore, $m_2$ is the unique $+$-based mass function of $\mu$ and can be built using the inductive algorithm sketched in Definition 19. Finally, the pseudo-product defined in Example 3 lacks an inverse (due to its behaviour for the value 0). As a consequence $m_3$, which*

*was also constructed following this algorithm, comes with no guarantee of uniqueness.*

It is worthwhile noticing that, when the capacity is decomposable according to $\boxplus$, the computation of its $\boxplus$-inverse transform is easy. We can indeed show that:

**Theorem 3.** *A capacity is $\boxplus$-decomposable iff it admits a $\boxplus$-based distribution.*

### 3.2 THE XEU DECISION RULE

Let us now extend the GEU decision rule (under the name XEU) in order to let it work on algebraic mass functions. To this extent, we first generalize the definition of expectation domains.

**Definition 20** (Extended expectation domain).
*An extended expectation domain is a tuple $E = (U^*, W^*, V^*, \boxplus, \mathbf{f}, \oplus, \otimes)$ where:*

- $U^*$ *is ordered by some reflexive $\succeq^{U^*}$;*
- $W^*$ *is ordered by some $\succeq^{W^*}$ which is reflexive, antisymmetric and transitive;*
- $V^*$ *is ordered by some reflexive $\succeq^{V^*}$;*
- $\boxplus : W^* \times W^* \to W^*$ *is an associative and commutative operator and admits a neutral element $0_\boxplus$;*
- $\mathbf{f} : \mathcal{MS}(U^*) \to U^*$ *is a function which aggregates multisets of utility degrees into a single utility value such that $\forall x, \mathbf{f}(\{x\}) = x$;[4]*
- $\oplus : V^* \times V^* \to V^*$ *is a commutative and associative operator;*
- $\otimes : W^* \times U^* \to V^*$.

This definition extends Definition 9 by adding to the framework the $\boxplus$ operator (in order to handle mass functions). Moreover, since a mass function involves *sets* of states, the focal elements, an act yields a *multiset* of utility degrees for each of them. The role of function $\mathbf{f}$ is to aggregate these utility degrees into a single one; of course, when the knowledge can be captured by a distribution, we must have $\mathbf{f}(\{x\}) = x$.

We can now set the definition of the XEU of an act:

**Definition 21** (XEU). *Let $\mathcal{D} = (\mathcal{A}, U, W, u, \mu)$ be a decision problem and $E = (U^*, W^*, V^*, \boxplus, \mathbf{f}, \oplus, \otimes)$ be an extended expectation domain such that $U \subseteq U^*$ and $W \subseteq W^*$. The XEU of an act $\chi$ of $\mathcal{D}$ by $E$ is:[5]*

---

[4]$\mathcal{MS}(U)$ denotes the set of all multisets in $U$ (i.e. subsets of $U$ with possibly multiple occurrences of the same element).

[5]When $\boxplus$ has no inverse, there may be several $\boxplus$-based mass functions of $\mu$, hence Definition 21 should be written:

$$\text{XEU}_\mathcal{D}^E(\chi) = \min_{m \in \mathcal{M}_\mu^\boxplus} \bigoplus_{B \subseteq \Omega} m(B) \otimes \mathbf{f}(u(\chi(B)))$$

where $\mathcal{M}_\mu^\boxplus$ is the set of all $\boxplus$-based mass functions of $\mu$. However,

$$\text{XEU}_\mathcal{D}^E(\chi) = \bigoplus_{B \subseteq \Omega} m_\mu^\boxplus(B) \otimes \mathbf{f}(u(\chi(B))),$$

*where $u(\chi(B)) = \{\{u(\chi(\omega)) \mid \omega \in B\}\}$ is the image of $B$ by $\chi$ and $u$, i.e. the multiset of utility degrees that $\chi$ reaches for some state of $B$.*

**Definition 22** (XEU-representation). *Let $\mathbf{v}$ be a decision rule and $E = (U^*, W^*, V^*, \boxplus, \mathbf{f}, \oplus, \otimes)$ be an extended expectation domain. $E$ is an XEU-representation of $\mathbf{v}$ iff for any decision problem $\mathcal{D} \in \text{Dom}(\mathbf{v})$ and any two acts $\chi$ and $\chi'$ of $\mathcal{D}$:*

$$\mathbf{v}(\chi) \succeq^\mathbf{v} \mathbf{v}(\chi') \iff \text{XEU}_\mathcal{D}^E(\chi) \succeq^{V^*} \text{XEU}_\mathcal{D}^E(\chi').$$

When $\mu$ is a probability measure, the EU expression is recovered as $E^{\text{EU}} = (\mathbb{R}, [0, 1], \mathbb{R}, +, \mathbf{f}, +, \times)$, where $\mathbf{f}$ is any function satisfying the condition $\mathbf{f}(\{x\}) = x$. In this context, function $\mathbf{f}$ is not directly involved.

Another series of examples is provided by decision rules tailored for situations of total ignorance, where the knowledge asserts that the real world lies within a subset $Q$ of $\Omega$ (and does not assert anything more). In other terms, the range of application of these rules is limited to capacities $\mu_Q$ defined by $\mu_Q(A) = 1$ if $Q \subseteq A$ and $\mu_Q(A) = 0$ otherwise. For the sake of simplicity, let $E = (\mathbb{R}, [0, 1], \mathbb{R}, +, \mathbf{f}, +, \times)$ be the expectation domain. We then derive $m_{\mu_Q}^+(Q) = 1$ and $m_{\mu_Q}^+(A) = 0$ for any $A \neq Q$; hence $\text{XEU}(\chi) = \mathbf{f}(u(\chi(Q)))$ for any act $\chi$.

Wald's rule [Wald, 1949] is recovered when $\mathbf{f}(X) = \min(X)$: $\mathbf{f}$ amounts to consider the worst-case utility. Laplace's rule is obtained by setting $\mathbf{f}(X) = \sum_{x \in X} x/|X|$: $\mathbf{f}$ captures an average utility approach. Hurwicz's rule [Hurwicz, 1951] is recovered when $\mathbf{f}(X) = \alpha \min(X) + (1 - \alpha) \max(X)$: $\mathbf{f}$ makes a trade off between optimistic and pessimistic attitudes towards ignorance.

More generally, when $\mu$ encodes a situation of total ignorance, i.e. involves an single focal element $Q$, $\text{XEU}_\mathcal{D}^E(\chi) = \mathbf{f}(u(\chi(Q)))$. Function $\mathbf{f}$ is the way to capture the behavior of the decision maker under ignorance. With the XEU rule, this principle is extended to sets $B$: $\mathbf{f}(u(\chi(B)))$ is the utility of $\chi$ when the DM knows that the real world is in $B$ and nothing more. We shall for instance set $\mathbf{f} = \min$, $\boxplus = \oplus = +$ and $\otimes = \times$: we will see in Section 4 that this domain captures the Choquet integral.

Before entering in the details of particular rules, let us establish the following representation theorem which shows that the only necessary condition for a rule to be representable

---

most of the rules proposed in the literature either lead to commutative groups, in which case the transform is unique, or can be expressed as Sugeno integrals, in which case the same value is provided whatever the max-based mass function considered. For the sake of readability, we omit the minimization in Definition 21.

as an XEU is that acts that lead to the same utility with the same beliefs have to be ordered in the same way – this condition is less demanding (and implied by) Chu and Halpern's condition of uniformity.

**Definition 23** (Restricted uniformity). *A decision rule* $\mathbf{v}$ *satisfies* restricted uniformity *iff for all* $\mathcal{D} \in \mathrm{Dom}(\mathbf{v})$ *and all acts* $\chi_1, \chi_2, \chi'_1, \chi'_2$ *of* $\mathcal{D}$ *such that for any subset* $B$ *of* $U$, $\mu(\{\omega \mid u(\chi_i(\omega)) \in B\}) = \mu(\{\omega \mid u(\chi'_i(\omega)) \in B\})$, *it holds that:*

$$\mathbf{v}_{\mathcal{D}}(\chi_1) \preceq^V \mathbf{v}_{\mathcal{D}}(\chi_2) \iff \mathbf{v}_{\mathcal{D}}(\chi'_1) \preceq^V \mathbf{v}_{\mathcal{D}}(\chi'_2).$$

**Theorem 4.** *Any decision rule that satisfies restricted uniformity admits an XEU-representation.*

Restricted uniformity weakens uniformity (Definition 13) in order to address both atomic utility degrees and sets of utility degrees. Notably, as uniformity—a prerequisite for a rule to have a GEU representation—implies restricted uniformity, decision rules which admit GEU representations also admit XEU representations.

**Theorem 5.** *Any decision rule which has a GEU-representation has an XEU-representation.*

Some rules satisfy restricted uniformity without being uniform—this is the case of many rules based on the Choquet integral. Hence XEU is more expressive than GEU.

# 4 EXACT XEU-REPRESENTATIONS OF DECISION RULES

Finally, in order to highlight the generality of the XEU rule, let us explore decision rules proposed in the literature. Because restricted uniformity is a very weak condition, XEU-representations exist for a large range of these rules. In the following we look for *exact* representations in the sense that the XEU computation provides the *same* score than the rule—this is important when the utility scores carry relevant information about the satisfaction of the DM (e.g. an expected gain or a guaranteed level of security).

**Definition 24** (Exact XEU-representation). *Let* $E = (U, W, V, \boxplus, \mathbf{f}, \oplus, \otimes)$ *be an extended expectation domain and* $\mathbf{v}$ *be decision rule compatible with* $U$ *and* $W$. $E$ *is an exact XEU-representation of* $\mathbf{v}$ *iff for any* $\mathcal{D} \in \mathrm{Dom}(\mathbf{v})$ *and any act* $\chi$ *of* $\mathcal{D}$:

$$\mathrm{XEU}_{\mathcal{D}}^E(\chi) = \mathbf{v}_{\mathcal{D}}(\chi).$$

That is, XEU assigns the same scores to acts than $\mathbf{v}$. Of course, any exact XEU-representation is an XEU-representation in the sense of Definition 22.

## 4.1 EXACT XEU-REPRESENTATIONS OF UNIFORM DECISION RULES

We know that a uniform decision rule does admit a GEU representation, and, thanks to Theorem 4, that it also admits an XEU representation. But an XEU and a GEU representation of the same rule are not necessarily exact representations of the rule, nor exact representations of each other in the sense that they do not necessarily provide the same value (but the same preference order on acts). What Theorem 4 guarantees is that if a GEU representation does exist for a rule, then an XEU representation can always be built for this rule. To get an equality between $\mathrm{XEU}_{\mathcal{D}}^{E^*}(\chi)$ and $\mathrm{GEU}_{\mathcal{D}}^E(\chi)$ (an exact representation of GEU by XEU based on the same $\oplus$ and $\otimes$ operators) we need a form of distributivity property:

**Theorem 6.** *Let* $E = (U, W, V, \oplus, \otimes)$ *be an expectation domain and* $E^* = (U, W, V, \boxplus, \min, \oplus, \otimes)$ *be an extended expectation domain which share the same domains, the same* $\oplus$ *operator and the same* $\otimes$ *operator. For any decision problem* $\mathcal{D} = (\mathcal{A}, U, W, u, \mu)$ *such that* $\mu$ *is* $\boxplus$-*decomposable, it holds that* $\mathrm{XEU}_{\mathcal{D}}^{E^*}(\chi) = \mathrm{GEU}_{\mathcal{D}}^E(\chi)$ *iff* $\forall w_1, w_2 \in W, \forall x \in U$:

$$(w_1 \otimes x) \oplus (w_2 \otimes x) = (w_1 \boxplus w_2) \otimes x.$$

Hence if $E = (U, W, V, \boxplus, \min, \oplus, \otimes)$ is an XEU-representation of a decision rule and if the distributivity-like condition holds, then $E' = (U, W, V, \oplus, \otimes)$ is a GEU-representation of it and both provide the same value. The condition obviously holds if $\oplus = \boxplus$ and $\otimes$ distributes over $\oplus$: if $\mu$ is $\oplus$-decomposable and $\otimes$ is distributive over $\oplus$ then for any $\chi$, $\mathrm{XEU}_{\mathcal{D}}^E(\chi) = \mathrm{GEU}_{\mathcal{D}}^E(\chi)$; i.e. the usual generalized expected utilities are XEUs.

It is worthwhile noticing that the equality between $\boxplus$ and $\oplus$ is not necessary. For instance, consider the possibilistic pessimistic decision rule proposed in [Dubois and Prade, 1995]. This rule works on a possibility measure $\Pi : 2^\Omega \to \Lambda$ and utility degrees in $\Lambda$ (scale $\Lambda = [0, 1]$ is classically used but any totally ordered scale can be handled as long as $\Lambda$ is equipped with a reverse operator $r : \Lambda \to \Lambda$, eg. $r(x) = 1 - x$ for $\Lambda = [0, 1]$):

$$\mathrm{U}^{\mathrm{Pess}}(\chi) = \min_{\omega \in \Omega} \, \max \Big( r(\Pi(\{\omega\})), \ u(\chi(\omega)) \Big).$$

Notice that the measure is decomposable (by $\max$), that its qualitative Möbius inverse $\gamma$ (Definition 15) is a distribution and that the rule is uniform. Letting $\mathrm{nmax}(x, y) = \max(r(x), y)$, $\mathrm{U}^{\mathrm{Pess}}$ can be rewritten twofold:

$$\mathrm{U}^{\mathrm{Pess}}(\chi) = \min_{x \in u(\chi(\Omega))} \mathrm{nmax} \Big( \Pi(\{\omega \mid u(\chi(\omega)) = x\}), x \Big),$$

$$\mathrm{U}^{\mathrm{Pess}}(\chi) = \min_{B \subseteq \Omega} \mathrm{nmax} \Big( \gamma(B), \max_{x \in u(\chi(B))} x \Big).$$

Hence, the possibilistic pessimistic decision rule has an exact GEU representation (from first equation):

$$E^{\mathrm{Pess}} = (\Lambda, \Lambda, \Lambda, \min, \mathrm{nmax})$$

and admits the following exact XEU-representation (from second equation) among others:

$$E^{\mathrm{Pess}'} = (\Lambda, \Lambda, \Lambda, \max, \max, \min, \mathrm{nmax}).$$

In this example $\boxplus = \max$ while $\oplus = \min$.

Among other uniform rules working with decomposable measures, that can thus be exactly captured by XEU and by GEU, let us cite expected utility (of course), the possibilistic optimistic rule proposed by [Dubois and Prade, 1995], and Wilson's order of magnitude decision rule [Wilson, 1995].

## 4.2 XEU-REPRESENTATIONS OF NON UNIFORM DECISION RULES

In order to highlight the generality of the XEU rule, let us now focus on some non uniform rules proposed in the literature—more precisely on rules that deal with non-decomposable measures. We show that even if they haven't any GEU representation, they have XEU representations, that are moreover exact.

**Choquet-like decision rules**

Choquet Expected Utility (CEU) is a general decision rule that can handle any kind of capacity ranging on the $[0, 1]$ interval, and in particular measures of lower probability [Gilboa and Schmeidler, 1994] (CEU thus captures the Ellsberg paradox [Ellsberg, 1961]), belief functions and of course probability measures. It also captures the rank-dependent utility rule [Quiggin, 1982] (that suits the Allais's paradox [Allais, 1953]) or the rule proposed by [De Campos et al., 1994] to handle probability intervals.

To any decision problem $\mathcal{D} = (\mathcal{A}, \mathbb{R}, [0, 1], u, \mu)$ and any act $\chi$ of $\mathcal{D}$, let us label the utility values reached by $\chi$ on $\Omega$ in such a way that $x_0^\chi \leq \cdots \leq x_n^\chi$. CEU is defined by:

$$\mathrm{CEU}_{\mathcal{D}}(\chi) = x_0^\chi + \sum_{i=1}^{n}(x_i^\chi - x_{i-1}^\chi)\mu(\{\omega \mid u(\chi(\omega)) \geq x_i^\chi\}).$$

**Proposition 2.**

$$\mathrm{CEU}_{\mathcal{D}}(\chi) = \sum_{B \subseteq \Omega} m_\mu^+(B) \times \min_{\omega \in B} u(\chi(\omega)),$$

It follows that the Choquet decision rule has an exact XEU representation:

**Proposition 3.** CEU *has an exact XEU-representation:*

$$E^{\mathrm{Choq}} = (\mathbb{R}, \mathbb{R}, \mathbb{R}, +, \min, +, \times);$$

Let us cite two other decision rules closely related to CEU—both dedicated to the Dempster-Shafer theory of evidence. In the Transferable Belief Model [Smets and Kennes, 1994] the idea is to derive a probability distribution from $m$ by applying the Laplace principle to each focal element (setting $p(\omega) = \sum_{B, \omega \in B} \frac{m(B)}{|B|}$) and to compute the expected utility w.r.t. this distribution. This is perfectly equivalent to compute, for each focal set, the mean of the series of values obtained on this set. Formally, it holds that:

**Proposition 4.** TBEU *has an exact XEU-representation:*

$$E^{\mathrm{TBM}} = (\mathbb{R}, \mathbb{R}, \mathbb{R}, +, \mathrm{mean}, +, \times);$$

*where* $\mathrm{mean} : \mathcal{MS}(\mathbb{R}) \to \mathbb{R}$ *is defined by* $\mathrm{mean}(X) = \sum_{x \in X} x/|X|$ *if* $X \neq \emptyset$ *and* $\mathrm{mean}(\emptyset)$ *yield an arbitrary value.*

The second rule, Jaffray's rule [Jaffray, 1989], deals with a family of parameters $\alpha : 2^U \to [0, 1]$. The $\alpha(X)$ are pessimism indexes in the sense of Hurwicz's but expressed in the context of each set $X$ of utility values – Hurwicz index is recovered when $\alpha$ is a constant function.

**Proposition 5.** *For any* $\alpha : 2^U \to [0, 1]$, $\mathrm{JEU}^\alpha$ *has an exact XEU-representation:*

$$E_\alpha^{\mathrm{Jaff}} = (\mathbb{R}, \mathbb{R}, \mathbb{R}, +, \mathrm{minmax}_\alpha, +, \times);$$

*where* $\mathrm{minmax}_\alpha : \mathcal{MS}(\mathbb{R}) \to \mathbb{R}$ *is defined by* $\mathrm{minmax}_\alpha(X) = \alpha(X) \min(X) + (1 - \alpha(X)) \max(X)$ *for all* $X \neq \emptyset$.

$E^{\mathrm{Choq}}$ is a special case of $E^{\mathrm{Jaff}}$ (letting $\alpha(X) = 1 \; \forall X$) - which coheres with the fact that the CEU rule, when applied to belief functions, is a special case of Jaffray's.

Those three expectation domains $E^{\mathrm{Choq}}$, $E^{\mathrm{TBM}}$ and $E^{\mathrm{Jaff}}$ share the same domains, orders and operators ($\boxplus = +$, $\otimes = \times$ and $\oplus = +$). They differ on the $\mathbf{f}$ function only. Of course, for singletons, $\mathbf{f}(\{x\}) = x$ in the three rules: we recover the fact that when $\mu$ is a probability measure, the three rules do coincide and simply come down to expected utility.

**Sugeno Decision Rules**

The Sugeno integral [Sugeno, 1974] is an ordinal counter part of the Choquet integral, based on $\max$ and $\min$ aggregations rather than on sums and products. Let $\mathcal{D} = (\mathcal{A}, \Lambda, \Lambda, u, \mu)$ be a decision problem where $\Lambda$ is totally ordered. The Sugeno value of an act is defined by:

$$\mathrm{SUG}_{\mathcal{D}}(\chi) = \max_{B \subseteq \Omega} \min\left(\mu(B), \min_{\omega \in B} u(\chi(\omega))\right).$$

Using the qualitative Möbius transform $\gamma$ one gets $\mathrm{SUG}_{\mathcal{D}}(\chi) = \max_{B \subseteq \Omega} \min\left(\gamma(B), \min_{\omega \in B} u(\chi(\omega))\right)$. It follows that SUG has an exact XEU representation.

**Proposition 6.** SUG *has an exact XEU-representation*

$$E^{\mathrm{Sugn}} = (\Lambda, \Lambda, \Lambda, \max, \min, \max, \min);$$

**Incomplete preference relations**

All the previous rules involve classical capacities, on the $[0, 1]$ interval, and yield complete orders of the acts, using a score in $\mathbb{R}$—they are summarized in Table 3, each rule deriving from the choice of a type of capacity (lines) and of an expectation domain (columns).

|  | $E^{\mathrm{Choq}}$ | $E^{\mathrm{TBM}}$ | $E^{\mathrm{Jaff}}$ | $E^{\mathrm{Sugn}}$ |
|---|---|---|---|---|
| $[0,1]$-capa | Choquet | – | – | SUG |
| Belief f. | CEU | TBEU | JEU | – |
| Proba. | EU | EU | EU | – |
| Possi. | CEU | – | – | $U^{\mathrm{Opt}}$ |
| Nec. | – | – | – | $U^{\mathrm{Pess}}$ |
| Total Ign. | Wald | Laplace | Hurwicz | Max |

Table 3: XEU-representations of common real-valued decision rules; the type of capacity used is specified in lines, the expectation domain in columns.

XEU shall also capture less classical rules, that do not produce complete relations. The first example is Wilson's order of magnitude decision rule [Wilson, 1995] which leads to a incomplete preference relation among acts—the extended domain simply relies on the domain $\mathbb{R}^o$ and on the operators $\oplus$ and $\otimes$ defined by Wilson (for the shake of brevity, we let the reader refer to [Wilson, 1995]) – we simply let $\boxplus = \oplus$ and, because the measure is decomposable, $\mathbf{f}$ can be any function such that $\mathbf{f}(\{x\}) = x$.

Another example is the decision rule proposed in [Denœux and Shenoy, 2020] for belief functions (here denoted as DSEU) – this rule is not uniform and thus cannot be captured by the GEU model. As for JEU, local pessimism indexes are considered. Indeed, DSEU involves two families of contextual indexes: $\alpha : 2^U \to [0, 1]$ and $\beta : 2^U \to [0, 1]$. DSEU considers real-valued utility functions and real-valued capacities (belief functions). However, unlike the rules presented in the previous sections, DSEU produces scores that are not real numbers, but intervals. In more details, let $[a, b]$ be the interval associated with some act $\chi$ and $[a', b']$ that of $\chi'$; then $\chi \succeq^{\mathrm{DSEU}} \chi' \iff (a \geq a') \wedge (b \geq b')$. Relation $\succeq^{\mathrm{DSEU}}$ is not complete in the general case. In order to cast DSEU in the XEU model, we will consider any real value $x$ as equal to the interval $[x, x]$: this allows us to properly define the function $\mathbf{f}_{\alpha,\beta}^{\mathrm{DSEU}}$.

**Proposition 7.** *For any* $\alpha : 2^U \to [0, 1]$ *and* $\beta : 2^U \to [0, 1]$ *such that* $\forall X, 0 \leq \alpha(X) \leq \beta(X) \leq 1$, $\mathrm{DSEU}_{\alpha,\beta}$ *has an exact XEU representation:*

$$E_{\alpha,\beta}^{\mathrm{DS}} = (\mathbb{R}, [0, 1], \mathbb{R}^2, +, \mathbf{f}_{\alpha,\beta}^{\mathrm{DS}}, \oplus^{\mathrm{DS}}, \otimes^{\mathrm{DS}});$$

*where:*

- $x \otimes^{\mathrm{DS}} [a, b] = [x \times a, x \times b]$;
- $[a, b] \oplus^{\mathrm{DS}} [a', b'] = [a + a', b + b']$;
- $\mathbf{f}_{\alpha,\beta}^{\mathrm{DS}}(X) = [\alpha(X) \min(X),$
$\alpha(X) \min(X) + \beta(X) \max(X)]$.

## 5 CONCLUSION

This paper has proposed a generalization of Chu and Halpern's GEU algebraic framework, enabling the representation of non uniform decision rules, and in particular of the Choquet and Sugeno integrals. The XEU formulation we propose puts forward the notion of algebraic mass functions as a way to capture elementary knowledge and highlights the use of a utility aggregator, the function $\mathbf{f}$, that captures the behavior of the decision maker when facing ignorance. This allows us for instance to compare at a glance several rules based on belief functions: the difference between the pignistic approach, the pessimistic integral, and the optimistic integral does not lay in the treatment of the knowledge (it's always the same knowledge, the mass function $m$), but in the compensatory/pessimistic/optimistic attitude: in the first case, $\mathbf{f}$ computes the mean value over each focal set, in the second, the decision-maker is cautious/robust and $\mathbf{f} = \min$ while in the third case, he/she would be (very) adventurous and $\mathbf{f} = \max$. From this, many variants of the Choquet integral can then be foreseen depending on the function $\mathbf{f}$ they may use, for instance a median or any other OWA. An orthogonal direction deriving from such an homogeneous approach is the design of interactive elicitation process that, through proposals for decision comparisons, allows the parameterization of function $\mathbf{f}$ (e.g. as done in [Adam and Destercke, 2021] for the elicitation of OWAs).

Algebraic approaches provide efficient frameworks to express general theoretical results, but also to specify the range of application of algorithms, as done e.g. in [Schiex et al., 1995, Pralet et al., 2007, Perny et al., 2005]. From a more practical point of view, our aim is to study optimization problems and to relate their tractability to the kind of information handled and to the way it is represented: a mass function may involve exponentially less focal elements that the measure. A complexity analysis and empirical measurements could help identify the benefit of algebraic mass functions and of XEU-representations in terms of computational efficiency.

From this point, the next step is of course to build a comprehensive axiomatics, in the sense of Savage in particular, which will derive the decision rule from the behavior expected when facing total ignorance—and to extend the XEU framework to the infinite case. In addition to its theoretical significance, such an extension may potentially broaden the applicability of XEU to domains where continuous or unbounded information is prevalent.

## Acknowledgements

Both authors have benefited from the AI Interdisciplinary Institute ANITI. ANITI is funded by the French "Investing for the Future – PIA3" program under the Grant agreement n°ANR-19-PI3A-0004.

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

# A  PROOFS FOR SECTION 2 (BACKGROUND AND MOTIVATIONS)

**Theorem 1** (Chu and Halpern [2004])**.** *A decision rule admits a GEU representation iff it is uniform and respects utility.*

*Proof.* From [Chu and Halpern, 2004, Theorem 3.5]. □

# B  PROOFS FOR SECTION 3 (GENERALIZING EXPECTED UTILITY – A STEP FORWARD )

**Proposition 1.** *$m$ is a $\boxplus$-based mass function of $\mu$ iff:*

$$\mu(\emptyset) = m(\emptyset), \quad and$$
$$\mu(A) = m(A) \boxplus \bigboxplus_{B \subsetneq A} m(B) \qquad\qquad \forall A \neq \emptyset.$$

*Proof.* It directly follows from Definition 16. □

**Theorem 2.** *When $(W^*, \boxplus)$ is a commutative group, $m_\mu^\boxplus$ is the unique $\boxplus$-mass function of $\mu$.*

*Proof.* By definition, $m_\mu^\boxplus$ exists and is a $\boxplus$-mass function of $\mu$. Uniqueness is shown by induction on the cardinality of $\mu$'s argument. Let $m$ be a $\boxplus$-based mass function of $\mu$. For $n = 0$, by definition, $m(\emptyset) = \mu(\emptyset) = m_\mu^\boxplus(\emptyset)$. Now, suppose that $m(B) = m_\mu^\boxplus(B)$ when $|B| < n$, $n > 0$. Then, for any $A$ such that $|A| = n$, we have $\mu(A) = m_\mu^\boxplus(A) + \bigboxplus_{B \subsetneq A} m_\mu^\boxplus(B) = m(A) + \bigboxplus_{B \subsetneq A} m(B)$, implying $m_\mu^\boxplus(A) = m(A)$ since $\boxplus$ has an inverse. By induction, $m_\mu^\boxplus(A) = m(A)$ for all $A$, i.e., $m_\mu^\boxplus$ is unique. □

**Theorem 3.** *A capacity is $\boxplus$-decomposable iff it admits a $\boxplus$-based distribution.*

*Proof.* To prove the theorem, we establish both the "if" and "only if" directions.

- *(if)* Assume that $\mu$ is a capacity admitting a distribution $m$. For any two disjoint events $A$ and $B$, it holds that $\mu(A \cup B) = \bigboxplus_{\omega \in A \cup B} m(\{\omega\}) = \mu(A) \boxplus \mu(B)$, hence $\mu$ is $\boxplus$-decomposable.
- *(only if)* Conversely, let $\mu$ be a $\boxplus$-decomposable capacity. From [Friedman and Halpern, 1995], $\mu(\emptyset) = \perp$ is neutral for $\boxplus$. Let $m : 2^\Omega \to W$ be the distribution such as $m(\{\omega\}) = \mu(\{\omega\})$ for all $\omega$ and $m(A) = \perp$ otherwise. Also, let $P(n) \iff (\forall |A| \leq n, \mu(A) = \bigboxplus_{\omega \in A} m(A))$ be the predicate that $m$ is a $\boxplus$-distribution of $\mu$ "up to sets of size $n$". Clearly, $P(0)$ and $P(1)$ hold. Now suppose $P(n)$ for $n > 0$ and $A \subseteq \Omega$ such that $|A| = n + 1$ and let $\omega^* \in A$. The sets $\{\omega^*\}$ and $A \setminus \{\omega^*\}$ are disjoint, their cardinalities are $\leq n$. From $\mu$ is decomposable, $\mu(A) = \mu(\{\omega^*\}) \boxplus \mu(A \setminus \{\omega^*\}) = m(\{\omega^*\}) \boxplus \bigboxplus_{\omega \in A \setminus \{\omega^*\}} m(\{\omega\}) = \bigboxplus_{\omega \in A} m(\{\omega\})$, hence $P(n + 1)$ holds. By induction, $m$ is a $\boxplus$-distribution of $\mu$. □

**Theorem 4.** *Any decision rule that satisfies restricted uniformity admits an XEU-representation.*

*Proof.* Let $\mathbf{v}$ be a decision rule that satisfies restricted uniformity. We first induce from $\mathbf{v}$ a relation $\succeq^M$ on "utility-based capacities". Using it, we then define a naive XEU representation in which all computations are deferred to $\succeq^V$ that finally compare acts. Lastly, we ensure this XEU representation is properly defined.

Let $\mathcal{D} \in \mathrm{Dom}(\mathbf{v})$ be a decision problem. For any act $\chi$ of $\mathcal{D}$, let $\mu_\chi : 2^U \to W$ denote the capacity induced from $\chi$ such that $\mu_\chi(X) = \mu(\{\omega \mid u(\chi(\omega)) \in X\})$. Hence it follows from $\mathbf{v}$'s restricted uniformity that $\forall \chi_1, \chi_1', \chi_2, \chi_2'$ such that $\forall i, \mu_{\chi_i} = \mu_{\chi_i'}$, we have $\mathbf{v}_\mathcal{D}(\chi_1) \succeq^{\mathbf{v}} \mathbf{v}_\mathcal{D}(\chi_2) \iff \mathbf{v}_\mathcal{D}(\chi_1') \succeq^{\mathbf{v}} \mathbf{v}_\mathcal{D}(\chi_2')$. That is, $\mathbf{v}$ orders acts with respect to the capacity they induce. In particular, for $\chi_1$ and $\chi_1'$ such that $\mu_{\chi_1} = \mu_{\chi_1'}$, it holds that $\mathbf{v}_\mathcal{D}(\chi_1) \sim^{\mathbf{v}} \mathbf{v}_\mathcal{D}(\chi_1')$ and $\forall \chi_2, \mathbf{v}_\mathcal{D}(\chi_1) \succeq^{\mathbf{v}} \mathbf{v}_\mathcal{D}(\chi_2) \iff \mathbf{v}_\mathcal{D}(\chi_1') \succeq^{\mathbf{v}} \mathbf{v}_\mathcal{D}(\chi_2)$. Hence one can induce the relation $\succeq^M$ on the set of capacities $M = \{\mu_\chi \mid \chi \in \mathbb{A}\}$ defined, for any acts $\chi_1$ and $\chi_2$, by $\mu_{\chi_1} \succeq^M \mu_{\chi_2} \iff \mathbf{v}_\mathcal{D}(\chi_1) \succeq^{\mathbf{v}} \mathbf{v}_\mathcal{D}(\chi_2)$. Note that $\succeq^M$ is well defined since we have shown that acts inducing the same capacity are equivalent and equally related to others acts.

Now, let $E = (U^*, W^*, V^*, \boxplus, \mathbf{f}, \oplus, \otimes)$ be the extended expectation such as:

- $U^* = 2^U \setminus \{\emptyset\}$ is the extension of $\mathbf{v}$'s utility domain (we equate values $x \in U$ to singletons $\{x\} \in U^*$). It is ordered by $\succeq^{U^*}$ such that $X \succeq^{U^*} Y \iff \min_{x \in X} x \succeq^U \min_{y \in Y} y$;
- $W^* = W$ is ordered by $\succeq^W$;
- $V^* = 2^{W^* \times U^*}$ is ordered by $\succeq^{V^*}$ such that

$$X \succeq^{V^*} Y \iff \left( C \mapsto \max_{\substack{(w,B) \in X \\ B \subseteq C}} w \right) \succeq^M \left( C \mapsto \max_{\substack{(w,B) \in Y \\ B \subseteq C}} w \right);$$

- $\boxplus = \max$;
- $\mathbf{f}$ is the identity function ($\forall B \in U^*, f(B) = B$);
- $\otimes : W \times U^* \to V^*$ is defined by $w \otimes X = \{(w,X)\}$;
- $\oplus : U^* \times U^* \to U^*$ is defined by $X \oplus Y = \{(w,B) \mid (w,B) \in X \cup Y \wedge w \neq \bot\}$.

We still have to check that $E$ is a proper extended expectation domain and that it defines an XEU representation of $\mathbf{v}$. First note that:

- $(W, \max, \bot)$ is a commutative monoid;
- $\forall x \in U, \mathbf{f}(\{x\}) = \{x\}$ (which is equated with $x$);
- $\oplus$ is a commutative and associative.

So $E = (U, W, V, \boxplus, \mathbf{f}, \oplus, \otimes)$ is a proper extended expectation domain. Finally, note that:

$$\begin{aligned}
\mathrm{XEU}_{\mathcal{D}}^E(\chi) &= \bigoplus_{B \subseteq \Omega} m_\mu^{\max}(B) \otimes \mathbf{f}(u(\chi(B))) \\
&= \bigoplus_{B \subseteq \Omega} \left\{ \left( m_\mu^{\max}(B), u(\chi(B)) \right) \right\} \\
&= \left\{ \left( m_\mu^{\max}(B), u(\chi(B)) \right) \mid B \subseteq \Omega \wedge m_\mu^{\max}(B) \neq \bot \right\}.
\end{aligned}$$

Hence for any $\mathcal{D}$ and any $\chi$ of $\mathcal{D}$:

$$\left( C \mapsto \max_{\substack{(w,B) \in \mathrm{XEU}_{\mathcal{D}}^E(\chi) \\ B \subseteq C}} w \right) = \left( C \mapsto \max_{\substack{(w,B) \in \mathrm{XEU}_{\mathcal{D}}^E(\chi) \\ B \subseteq C}} m_\mu^{\max}(B) \right) = \mu_\chi$$

It then follows that $\mathrm{XEU}_{\mathcal{D}}^E(\chi) \succeq^{V^*} \mathrm{XEU}_{\mathcal{D}}^E(\chi') \iff \mu_\chi \succeq^M \mu_{\chi'} \iff \mathbf{v}_{\mathcal{D}}(\chi) \succeq^{\mathbf{v}} \mathbf{v}_{\mathcal{D}}(\chi')$, hence $E$ is an XEU representation of $\mathbf{v}$. $\qquad\square$

**Theorem 5.** *Any decision rule which has a GEU-representation has an XEU-representation.*

*Proof.* We first show that uniformity implies restricted uniformity.

Let $\mathbf{v}$ be a uniform decision rule. For any decision problem $\mathcal{D} \in \mathrm{Dom}(\mathbf{v})$ and for all acts $\chi_1, \chi_1', \chi_2, \chi_2'$ of $\mathcal{D}$ such that $\mu(\{\omega \mid u(\chi_i(\omega)) = x\}) = \mu(\{\omega \mid u(\chi_i'(\omega)) = x\})$ for all $x \in U$ and $i = 1, 2$, it holds that:

$$\mathbf{v}(\chi_1) \succeq \mathbf{v}(\chi_2) \iff \mathbf{v}(\chi_1') \succeq \mathbf{v}(\chi_2').$$

Now, suppose four acts $\chi_3, \chi_3', \chi_4, \chi_4'$ of $\mathcal{D}$ such that $\mu(\{\omega \mid u(\chi_i(\omega)) \in X\}) = \mu(\{\omega \mid u(\chi_i'i(\omega)) \in X\})$ for all $X \subseteq U$ and $i = 3, 4$. Since this equality holds for any singleton $X = \{x\}$ ($x \in U$), one can rewrite $\mu(\{\omega \mid u(\chi_i(\omega)) = x\}) = \mu(\{\omega \mid u(\chi_i'i(\omega)) = x\})$ for $i = 3, 4$. Hence, from $\mathbf{v}$'s uniformity, we have $\mathbf{v}(\chi_3) \succeq \mathbf{v}(\chi_4) \iff \mathbf{v}(\chi_3') \succeq \mathbf{v}(\chi_4')$, that is, uniformity implies restricted uniformity.

The proof is now straightforward: suppose $\mathbf{v}$ has a GEU representation; hence it satisfies uniformity (from Theorem 1), hence it satisfies restricted uniformity, hence it has an XEU representation (from Theorem 4). $\qquad\square$

## C PROOFS FOR SECTION 4 (EXACT XEU-REPRESENTATIONS OF DECISION RULES)

**Theorem 6.** *Let $E = (U, W, V, \oplus, \otimes)$ be an expectation domain and $E^* = (U, W, V, \boxplus, \min, \oplus, \otimes)$ be an extended expectation domain which share the same domains, the same $\oplus$ operator and the same $\otimes$ operator. For any decision problem $\mathcal{D} = (\mathcal{A}, U, W, u, \mu)$ such that $\mu$ is $\boxplus$-decomposable, it holds that $\text{XEU}_{\mathcal{D}}^{E^*}(\chi) = \text{GEU}_{\mathcal{D}}^E(\chi)$ iff $\forall w_1, w_2 \in W, \forall x \in U$:*

$$(w_1 \otimes x) \oplus (w_2 \otimes x) = (w_1 \boxplus w_2) \otimes x.$$

*Proof.* Firstly, note that according to [Friedman and Halpern, 1995], $\boxplus$ have a neutral element $0_\boxplus$, and according to Theorem 2, $m_\mu^\boxplus$ exists and is unique.

- *(if)* Let $w_2 = 0_\boxplus$. For any $w_1$ and $x$, $(w_1 \otimes x) \oplus (0_\boxplus \otimes x) = (w_1 \boxplus 0_\boxplus) \otimes x = (w_1 \otimes x)$. Let $\chi$ be an act of $\mathcal{D}$ and let $P_x(B) = (x = \min_{\omega \in B} u(\chi(\omega)))$. Since $u \circ \chi$ defines a partition of $\Omega$, it follows that $\text{GEU}_{\mathcal{D}}^E(\chi) = \bigoplus_{x \in \text{Img}(u \circ \chi)} \mu(\{\omega \mid P_x(\{\omega\})\}) \otimes x = \bigoplus_{x \in \text{Img}(u \circ \chi)} \left( \boxplus_{\omega \in \Omega, P_x(\{\omega\})} m_\mu^\boxplus(\{\omega\}) \right) = \bigoplus_{\omega \in \Omega} m_\mu^\boxplus(B) \otimes u(\chi(\omega))$. Then, since $\oplus$ is associative and commutative, one can rewrite $\text{XEU}_{\mathcal{D}}^E(\chi) = \bigoplus_{x \in \text{Img}(u \circ \chi)} \bigoplus_{B \subseteq \Omega, P_x(B)} m_\mu^\boxplus(B) \otimes x$. Lastly, for any $x \in \text{Img}(u \circ \chi)$, one may split the sum for $|B| = 1$ and $|B| \neq 1$. Since $|B| \neq 1 \implies m_\mu^\boxplus(B) = 0_\boxplus$, we have $\bigoplus_{B \subseteq \Omega, p_x(B)} m_\mu^\boxplus(B) \otimes x = \left( \bigoplus_{B \subseteq \Omega, p_x(B) \wedge |B|=1} m_\mu^\boxplus(B) \otimes x \right) \oplus (0_\boxplus \otimes x) = \bigoplus_{\omega \in \Omega, P_x(\{\omega\})} m_\mu^\boxplus(\{\omega\})$. Hence $\text{XEU}_{\mathcal{D}}^E(\chi) = \text{GEU}_{\mathcal{D}}^E(\chi)$ for any act $\chi$ of $\mathcal{D}$.

- *(only if)* Suppose $w_1, w_2 \in W$ and $x \in U$ such that $(w_1 \otimes x) \oplus (w_2 \otimes x) \neq (w_1 \boxplus w_2) \otimes x$. Let $\mathcal{D} = (\mathcal{A}, (U, W), u, \mu)$ be a decision problem where $\Omega = \{\omega_1, \omega_2\}$, where an act $\chi$ of $\mathcal{D}$ is such that $u(\chi(\omega_1)) = u(\chi(\omega_2)) = x$ and where $\mu(\{\omega_1\}) = w_1$ and $\mu(\{\omega_2\}) = w_2$. It follows that $\text{GEU}_{\mathcal{D}}^E(\chi) = \mu(\{\omega \mid P_x(\{\omega\})\}) \otimes x = \left( m_\mu^\boxplus(\{\omega_1\}) \boxplus m_\mu^\boxplus(\{\omega_2\}) \right) \otimes x \neq \left( m_\mu^\boxplus(\{\omega_1\}) \otimes x \right) \boxplus \left( m_\mu^\boxplus(\{\omega_2\}) \right) \otimes x = \text{XEU}_{\mathcal{D}}^E(\chi)$.

$\square$

**Proposition 2.**

$$\text{CEU}_{\mathcal{D}}(\chi) = \sum_{B \subseteq \Omega} m_\mu^+(B) \times \min_{\omega \in B} u(\chi(\omega)),$$

*Proof.* The proof is quite simple but requires many rewritings. Let $\mathcal{D} = (\mathcal{A}, (\mathbb{R}, [0,1]), u, \mu)$ be a decision problem and $\chi$ an act of $\mathcal{D}$. Then let $x_1 < \cdots < x_n$ be the ordered utility level reached by $\chi$ and let $A_i = \{\omega \mid u(\chi(\omega)) \geq i\}$ denote the set of states leading to a utility greater of equal to $x_i$. Notice that $\mu(A_0) = 1$ and let $\mu(A_{n+1}) = 0$. Hence:

$$
\begin{aligned}
\text{CEU}_{\mathcal{D}}(\chi) &= x_0 + (x_1 - x_0)\mu(A_1) + (x_2 - x_1)\mu(A_2) + \cdots + (x_n - x_{n-1})\mu(A_n) \\
&= x_0\mu(A_0) + x_1\mu(A_1) - x_0\mu(A_1) + x_2\mu(A_2) - x_1\mu(A_2) + \cdots + x_n\mu(A_n) - x_{n-1}\mu(A_n) \\
&= x_0\big(\mu(A_0) - \mu(A_1)\big) + x_1\big(\mu(A_1) - \mu(A_2)\big) + \cdots + x_n\big(\mu(A_n) - \mu(A_{n+1})\big) \\
&= \sum_{i=0}^{n} x_i\big(\mu(A_i) - \mu(A_{i+1})\big).
\end{aligned}
$$

Now, notice that $\forall i, A_{i+1} \subsetneq A_i$. Consider a set $B \subseteq A_i$. Any $\omega \in B$ leads to a utility $u(\chi(\omega)) \geq x_i$. It may or may not be an $\omega \in B$ that leads to $x_i$ exactly. Let $P_i$ denote this property, such that $P_i(B) \iff (B \subseteq A_i \wedge \exists \omega \in B, u(\chi(\omega)) = x_i)$. Hence by definition, $\forall B \subseteq \Omega, P_i(B) \iff \min_{\omega \in B} u(\chi(\omega)) = x_i$. Moreover, it also holds that $P_i(B) \iff (B \subseteq A_i \wedge B \nsubseteq A_{i+1})$. Hence:

$$
\begin{aligned}
\mu(A_i) - \mu(A_{i+1}) &= \left( \sum_{B \subseteq A_i} m(B) \right) - \left( \sum_{B \subseteq A_{i+1}} m(B) \right) \\
&= \left( \sum_{B \ s.t. P_i(B)} m(B) \right) + \left( \sum_{B \subseteq A_{i+1}} m(B) - m(B) \right) = \sum_{B \ s.t. P_i(B)} m(B).
\end{aligned}
$$

Last, notice that there is a single $i$ such that $P_i(B)$ for any $B \subseteq \Omega$. It thus defines a partition of $2^\Omega$. Hence one can rewrite:

$$\text{CEU}_\mathcal{D}(\chi) = \sum_{i=0}^{n} x_i \big(\mu(A_i) - \mu(A_{i+1})\big)$$

$$= \sum_{i=0}^{n} x_i \times \sum_{B \ s.t. \ P_i(B)} m(B)$$

$$= \sum_{i=0}^{n} \sum_{B \ s.t. P_i(B)} m(B) \times \min_{\omega \in B} u(\chi(B))$$

$$= \sum_{B \subseteq \Omega} m(B) \times \min_{\omega \in B} u(\chi(B)).$$

$\square$

**Proposition 3.** CEU *has an exact XEU-representation:*

$$E^{\text{Choq}} = (\mathbb{R}, \mathbb{R}, \mathbb{R}, +, \min, +, \times);$$

*Proof.* Direct from CEU's expression on the Möbius inverse, since $(\mathbb{R}, +, \times, 0, 1)$ is a semiring (with $\bot = 0_\boxplus = 0$) and $\forall x, \min(\{x\}) = x$. $\square$

**Proposition 4.** TBEU *has an exact XEU-representation:*

$$E^{\text{TBM}} = (\mathbb{R}, \mathbb{R}, \mathbb{R}, +, \text{mean}, +, \times);$$

*where* $\text{mean} : \mathcal{MS}(\mathbb{R}) \to \mathbb{R}$ *is defined by* $\text{mean}(X) = \sum_{x \in X} x/|X|$ *if* $X \neq \emptyset$ *and* $\text{mean}(\emptyset)$ *yield an arbitrary value.*

*Proof.* From its expression over the recovered probability distribution $\text{BetP}_\mu$ defined by $\text{BetP}_\mu(\omega) = \sum_{\omega \in B \subseteq \Omega} m_\mu^+(B)/|B|$, it holds that:

$$\text{TBEU}_\mathcal{D}(\chi) = \sum_{\omega \in \Omega} \text{BetP}_\mu(\omega) \times u(\chi(\omega)) = \sum_{B \subseteq \Omega} m_\mu^+(B) \times \sum_{\omega \in B} u(\chi(\omega))/|B|.$$

Furthermore, $(\mathbb{R}, +, \times, 0, 1)$ is a semiring (with $\bot = 0_\boxplus = 0$) and $\forall x, \text{mean}(\{x\}) = x$. Hence $E^{\text{TBM}}$ is an exact XEU representation of TBEU. $\square$

**Proposition 5.** *For any* $\alpha : 2^U \to [0,1]$, $\text{JEU}^\alpha$ *has an exact XEU-representation:*

$$E_\alpha^{\text{Jaff}} = (\mathbb{R}, \mathbb{R}, \mathbb{R}, +, \text{minmax}_\alpha, +, \times);$$

*where* $\text{minmax}_\alpha : \mathcal{MS}(\mathbb{R}) \to \mathbb{R}$ *is defined by* $\text{minmax}_\alpha(X) = \alpha(X) \min(X) + (1 - \alpha(X)) \max(X)$ *for all* $X \neq \emptyset$.

*Proof.* Direct from Jaffray's rule's expression, since $(\mathbb{R}, +, \times, 0, 1)$ is a semiring (with $\bot = 0_\boxplus = 0$) and $\forall x, \text{minmax}_\alpha(\{x\}) = x$. $\square$

**Proposition 6.** SUG *has an exact XEU-representation*

$$E^{\text{Sugn}} = (\Lambda, \Lambda, \Lambda, \max, \min, \max, \min);$$

*Proof.* From Sugeno's integral on the capacity: $U_\mathcal{D}^{\text{Sugn}}(\chi) = \max_{B \subseteq \Omega} \min(\mu(B), \min_{\omega \in B} u(\chi(\omega)))$, by noting that any max-based mass function is such that $m_\mu^{\max}(B) = \mu(B)$ or $m_\mu^{\max}(B) < \mu(B)$ (in this latter case, there exists $B' \subsetneq B$ such that $m_\mu^{\max}(B') = \mu(B)$). Iterating over all $B \subseteq \Omega$, one will consider either $m_\mu^{\max}(B) = \mu(B)$ and the minimal utility $x_B$ reachable in $B$ or $m_\mu^{\max}(B') = \mu(B)$ and the minimal utility $x_{B'}$ reachable in $B'$. Since $x_{B'} \succeq^U x_B$, it holds that $\max\big(\min(\mu(B), x_B), \min(\mu(B), x_{B'})\big) = \min(\mu(B), x_{B'})$, so $U_\mathcal{D}^{\text{Sugn}}(\chi) = \text{XEU}_{\mathcal{D}'}(\chi)$.

Furthermore, $(V, \max, \min, 0, 1)$ is a semiring (with $\bot = 0_\boxplus = 0$) hence $E^{\text{Sugn}}$ is an exact XEU representation of SUG. $\square$

**Proposition 7.** *For any $\alpha : 2^U \to [0,1]$ and $\beta : 2^U \to [0,1]$ such that $\forall X, 0 \le \alpha(X) \le \beta(X) \le 1$, $\mathrm{DSEU}_{\alpha,\beta}$ has an exact XEU representation:*

$$E^{\mathrm{DS}}_{\alpha,\beta} = (\mathbb{R}, [0,1], \mathbb{R}^2, +, \mathbf{f}^{\mathrm{DS}}_{\alpha,\beta}, \oplus^{\mathrm{DS}}, \otimes^{\mathrm{DS}});$$

*where:*

- $x \otimes^{\mathrm{DS}} [a,b] = [x \times a, x \times b]$;
- $[a,b] \oplus^{\mathrm{DS}} [a',b'] = [a + a', b + b']$;
- $\mathbf{f}^{\mathrm{DS}}_{\alpha,\beta}(X) = [\alpha(X)\min(X),$
  $\qquad\qquad \alpha(X)\min(X) + \beta(X)\max(X)]$.

*Proof.* From [Denœux and Shenoy, 2020], we obtain the above definitions so it only remains to check that $\succeq^{\mathrm{DSEU}}$ is reflexive, antisymmetric and transitive. Recall that $[a,b] \succeq^{\mathrm{DSEU}} [a',b'] \iff (a \ge a') \wedge (b \ge b')$, so $\succeq^{\mathrm{DSEU}}$ is obviously reflexive, transitive and antisymmetric since $\ge$ also is. Hence $E^{\mathrm{DS}}$ is a proper XEU-representation of DSEU. $\qquad\square$