# OpenReview forum: "Generalized Expected Utility as a Universal Decision Rule -- A Step Forward"
_auai.org/UAI/2024/Conference — UAI 2024 oral_

### Official Review · Reviewer_tPcs · 2024-03-19

**Q2-1 Originality-Novelty:** 3
**Q2-2 Correctness-Technical Quality:** 3
**Q2-5 Clarity Of Writing:** 3

**Q1 Summary And Contributions:**

The paper proposes a uniform algebraic framework to represent generalized expected utility (GEU) for different uncertainty measures. It extends previous GEU framework to cover a larger scope of decision rules, including those based on non-decomposable uncertainty measures such as Dempster-Shafer theory, Choquet integral, and Sugeno integral. The main results include the sufficient condition for the representability of decision rules in the proposed XEU framework, and sufficient and necessary condition for the exact representation of decision rules based on decomposable measures.

**Q2-3 Extent To Which Claims Are Supported By Evidence:**

3: Good: the main claims are supported by convincing evidence (in the form of adequate experimental evaluation, proofs, (pseudo-)code, references, assumptions).

**Q2-4 Reproducibility:**

4: Excellent: key resources (e.g. proofs, code, data) are available and key details (e.g. proof sketches, experimental setup) are comprehensively described for competent researchers to confidently and easily reproduce the main results.

**Q3 Main Strengths:**

A nice presentation of a theoretical framework based on novel ideas.

**Q4 Main Weakness:**

The significance of the contribution is less clear. It is unclear how such a framework can help decision makers to use the decision rules.

**Q5 Detailed Comments To The Authors:**

Because of the generality of the algebraic capacity and its corresponding mass function used in the proposed XEU framework, it can cover a lot of existing decision rules. On one hand, this is indeed a nice result and represents theoretical advance on the endeavor of integrating diverse uncertain formalisms. On the other hand, it is less clear how the work can have impact on the practice of decision-making under uncertainty. There is a short remark in the conclusion regarding advantages of the algebraic framework on the computational aspect of the decision rules. However, it need further elaboration on how easily algorithms developed for general algebraic structures can be transferred to more concrete domains. In addition, the fact that the framework can instantiate a diverse of existing decision rules does not provide much insight on the development of new decision theories. A more insightful direction is perhaps to find new instances of algebraic capacity or mass functions with practical sense and show that the framework can help design meaningful decision rules in those cases.

As there is no guarantee about the unique existence of \boxplus-mass function for a given capacity and as a result, it seems that m_{\mu}^\Boxplus is defined only for commutative group operator \boxplus (Def.19), it is not sure if the restriction is also applied to the definition of XEU since m_{\mu}^\Boxplus  explicitly appears in its formulation. If it is the case, then obviously, the applicability of XEU become more restrictive as it excludes many qualitative uncertainty measures. If not, does it mean that XEU may not exist or be not unique for any decision problem and expectation domain?

Below, I list some minor technical issues (or typos) to be clarified.

1. Table 1, \Pi violates monotony on {\omega_1} and {\omega1,\omega_3}

2. Def.10, typo on the expression for GEU: , :\mid

3. Def.12: as x_1 and x_2 are acts, the RHS should use u(x_1) and u(x_2)

4. Def.17: W or W^*?

5. L.2 after Def.18: identity element or neutral element?

6. Def.23: It seems that B is a subset of U?

7. Def.24: Is it implicitly assumed that W^*=W and U^*=U in the definition?

8. L.4, Sec.4.1: do not are not

9. Does the \boxplus-decomposability of a capacity imply the uniqueness of its  \boxplus-based mass function? Otherwise, the theorem may not hold for all of its \boxplus-based mass functions.

10. In rewriting U^{Pess}, it seems that the definition of nmax presupposes the existence of the minus operator in the domain \Lambda. When \Lambda is not [0,1], is the restriction reasonable?

11. In the definition of CEU, x_0\chi seems not defined.

12. The section entitled Partial Orderings, L.3: they are?

13. Proposition 7: it seems that the definition does not match with the XEU framework as U=\mathbb{R}, W=[0,1], and V=\mathbb{R}^2. According to Def.20, it should be \otimes: \mathbb{R}\times[0,1]\rightarrow\mathbb{R}^2 and {\bf f}:{\cal MS}(\mathbb{R})\rightarrow\mathbb{R}, but arguments of these two operators in the definition do not have correct types. It seems that changing U into \mathbb{R}^2 can partially address the issue (only for the \otimes operator). However, because it is said that DSEU considers real-valued utilit functions, I am not sure if the change complies with the original DSEU.

**Q9 Complying With Reviewing Instructions:**

Yes

---

> ### Author Rebuttal · Authors · 2024-04-03
>
> Thank you very very much for the detailed review and comments
>
>
> * Indeed, the work presented is essentially theoretical and is not aimed at an end user who would seek to make a decision in a practical case.  Regarding practical  perspectives the one we target is  primarily algorithmic (defining a generic dynamic programming algorithm).
>
> On the other hand, this work of general formalization can help the decision-makers "parameterize" a rule  not clearly defined at the begining of the decision processs. One can imagine, in an elicitation process, an interactive algorithm that, through proposals for decision comparisons, allows the parameterization of the function f, seen as an OWA (e.g. as done in Loïc Adam, Sébastien Destercke:
> Incremental Elicitation of Preferences: Optimist or Pessimist? ADT 2021: 71-85).
>
> * Yes, a first direction could be to work on the capacity and imagine new ones. Alternatively, we can also work on function $f$. The  presence of function f is one of the most interesting aspects of the formulation we propose (it it not appear in Chu and Halpern's GEU).
>
> For example, in the Choquet integral, one can see that the difference between the pignistic approach, the pessimistic integral, or the optimistic integral lies not in the treatment of knowledge (it's always the same knowledge, mm), but in the neutral/optimistic/pessimistic attitude: in the first case, f takes computes the mean value over each focal set, in the second, the decision-maker is cautious/robust (f=min), while in the third case, he/she would be adventurous (f=max).
>
> From this, one could then imagine a series of other rules that would vary f. For example, a rule of minimax regret per focal element, or the use of an OWA (ordered Weighted Average).
>
> However, the practical interest of these new rules and their properties would still need to be studied  and axiomatized  - the general algebraic framework could be a basis for such characterizations
>
>
> * Concerning the more technical questions:
>
> - Does the \boxplus-decomposability of a capacity imply the uniqueness of its \boxplus-based mass function? Otherwise, the theorem may not hold for all of its \boxplus-based mass functions.
> R: Yes, indeed  Friedman Halpern  show that "decomposable -> existence of a neutral element",  and we show that  "existence of a neutral element -> unicity".
>
> -  Def.24: Is it implicitly assumed that W^*=W and U^*=U in the definition?
> R: no
>
> - In rewriting U^{Pess}, it seems that the definition of nmax presupposes the existence of the minus operator in the domain \Lambda. When \Lambda is not [0,1], is the restriction reasonable?
> R: Yes, indeed.
>
> - Table 1, \Pi violates monotony on {\omega_1} and {\omega1,\omega_3}
> R: There is an error in the example, sorry (and thanks)
>
> - Proposition 7: (...)  It seems that changing U into \mathbb{R}^2 can partially address the issue (only for the \otimes operator). R: However, because it is said that DSEU considers real-valued utilit functions, I am not sure if the change complies with the original DSEU.
> R: Yes, we have that it mind - we thing about changing U into \mathbb{R}^2 and it  would comply with DSEU
>
> (the other points are mainly  typos that we have to correct - sorry - and thanks again)

---

### Official Review · Reviewer_qrGA · 2024-03-21

**Q2-1 Originality-Novelty:** 3
**Q2-2 Correctness-Technical Quality:** 3
**Q2-5 Clarity Of Writing:** 2

**Q10 Ethical Concerns:**

No.

**Q1 Summary And Contributions:**

This paper extends the generalized expected utility rule developed for decomposable capacities to a more general framework where the capacities are not necessarily decomposable. This general framework includes the GEU, and particular cases such as the Choquet expected utility rule or the Sugeno rule, among others.

**Q2-3 Extent To Which Claims Are Supported By Evidence:**

3: Good: the main claims are supported by convincing evidence (in the form of adequate experimental evaluation, proofs, (pseudo-)code, references, assumptions).

**Q2-4 Reproducibility:**

3: Good: key resources (e.g. proofs, code, data) are available and key details (e.g. proofs, experimental setup) are sufficiently well-described for competent researchers to confidently reproduce the main results.

**Q3 Main Strengths:**

The contribution deals with an interesting topic, generalizes existing results related to the expected utility representation with non-additive measures and the results seem correct and sound.

**Q4 Main Weakness:**

If I had to nitpick, I miss a deeper discussion about the role of the function f in the XEU. It is true that the authors discuss its relevance at the beginning of page 6-column 1, but I would suggest to expand this discussion because it would allow the reader to better understand its meaning. As well, I would appreciate if the authors add a (non-trivial) example in Section 3 showing the generality of the presented framework. Finally, I would suggest to thoroughly reviewing the writing.

**Q5 Detailed Comments To The Authors:**

Main comments: the contribution deals with an interesting topic, generalizes existing results related to the expected utility representation with non-additive measures and the results seem correct and sound.
If I had to nitpick, I miss a deeper discussion about the role of the function f in the XEU. It is true that the authors discuss its relevance at the beginning of page 6-column 1, but I would suggest to expand this discussion because it would allow the reader to better understand its meaning. As well, I would appreciate if the authors add a (non-trivial) example in Section 3 showing the generality of the presented framework. Finally, I would suggest to thoroughly reviewing the writing.
Evaluation: Anyway, apart from the comments above, the manuscript perfectly fits with the scope of the conference and I the content is interesting, hence I recommend to accept it.
Detailed comments (P=page, C=columna, L=line):
-Moebius -> M\” obius
-monotonic -> monotone
-P1,C2,L10: “…work of Halpern and al.”: do you mean Halpern at al? Please, add reference.
-Def 2: monotony -> monotonicity. Also: shouldn’t you detailed monotone increasing?
-Example 1: you may specify that you are using a probability measure, a belief function and a possibility measure.
-Def 10, last equation: shouldn’t \succeq be \succeq^v?
-Def 12 and 13: I suggest to write \succeq instead of \preceq to be consistent with the previous notation.
-Thm.1: please, add reference.
-P4, C1, L24: “monotonic” -> “monotonicity”.
-P4, C1, L-8: “regards” -> “regard”
-P4, C2, L9: “such as” -> “such that”. Also, add a final dot at the end of the itemize.
-P4, C2, sentence before def.18: it is unclear; please, rewrite.
-P5, C1, L4-5: unclear sentence; please, rewrite.
-Def.21: “such as” -> “such that”.
-P6, C1, paragraphs 1, 2 and 3: I would expand and detail this part to emphasize the relevance of the function f.
-P6, C1, L22: “show” -> “shows”.

**Q9 Complying With Reviewing Instructions:**

Yes

---

> ### Author Rebuttal · Authors · 2024-04-03
>
> Thank you very very much for the detailed review and comments
>
>
> About the function f indeed, its highlighting is one of the most interesting aspects of the formulation we propose based on the mass function. It does not appear at all in Chu and Halpern's formulation.
>
> For example, in the Choquet integral, one can see that the difference between the pignistic approach, the pessimistic integral, or the optimistic integral lies not in the treatment of knowledge (it's always the same knowledge, mm), but in the neutral/optimistic/pessimistic attitude: in the first case,  f takes computes the mean value over each focal set, in the second, the decision-maker is cautious/robust (f=min), while in the third case, he/she would be adventurous (f=max).
>
> From this, one could then imagine a series of other rules that would vary f. For example, a rule of minimax regret per focal element, or the use of an OWA (ordered Weighted Average). However, the practical interest of these new rules and their properties would still need to be studied.
>
> Yes, indeed, as you and another reviewer suggest, it is necessary to add a running example - to illustrate the rules and their formalization (we would consider an example based on the Ellsberg paradox, which highlights the differences between the rules).
>
> Thank you also very much for the detailed comments, which will allow us to improve the quality and readability of the article.

---

### Official Review · Reviewer_sreq · 2024-03-22

**Q2-1 Originality-Novelty:** 3
**Q2-2 Correctness-Technical Quality:** 3
**Q2-5 Clarity Of Writing:** 3

**Q1 Summary And Contributions:**

In order to capture a larger range of decision rules, the authors propose a generalization of Chu and Halpern's GEU algebraic framework, which allows the representation of non-consistent decision rules, in particular Choquet and Sugeno integrals.

**Q2-3 Extent To Which Claims Are Supported By Evidence:**

2: Fair: the main claims are somewhat supported by evidence (but the experimental evaluation may be weak, or does not match entirely with the claims, important baselines may be missing, proofs contain important ideas but lack rigor, algorithmic details are only discussed superficially, references are imprecise, assumptions are not sufficiently motivated or explicated, etc.).

**Q2-4 Reproducibility:**

3: Good: key resources (e.g. proofs, code, data) are available and key details (e.g. proofs, experimental setup) are sufficiently well-described for competent researchers to confidently reproduce the main results.

**Q3 Main Strengths:**

1)The paper provides related theoretical proof.
2)The topic of this paper belongs to the forefront of this field.
3)The explanation of relevant work is basically fine.

**Q4 Main Weakness:**

1)The paper does not give experiments or examples.
2)The paper lacks the prospect of future work.

**Q5 Detailed Comments To The Authors:**

1)The paper does not give experiments or examples.
2)The paper lacks the prospect of future work.
3)How does the writer argue that his work is better.

**Q9 Complying With Reviewing Instructions:**

Yes

---

> ### Author Rebuttal · Authors · 2024-04-03
>
> Thank you very much for the review.
>
> 1)The paper does not give experiments or examples.
>
> A: Adding experiments is challenging due to space limitations (and also because the paper, in its current version, is essentially theoretical), but it would be a future development worth considering.
>
> However, we should be able to add to the current paper a running example to illustrate the rules and their formalization (for example, based on the Ellsberg paradox).
>
>  2)The paper lacks the prospect of future work.
>
> A: Regarding  the perspectives, they would, for the practical part, be primarily algorithmic (defining a generic dynamic programming algorithm). From a theoretical standpoint, the objective is to define a unified axiomatization of the different existing rules, based on the XEU framework: which property guides the XEU towards one rule or another.
>
> The second theoretical development includes the extension to the infinite case (and studying the impact of the assumption of finiteness).
>
> We will try to put more emphasis on these points in the conclusion.
>
> 3)How does the writer argue that his work is better.
>
> A: The major argument of our work, compared to Chu and Halpern's previous results, is to be able to capture decision rules based on Choquet and Sugeno integrals, and more generally rules adapted to non-additive measures. It should also be noted that the XEU components simplify the modeling and definition of new decision rules by separating, in particular, the treatment of knowledge (e.g. variability) by the oplus operator and the treatment of ignorance by the f function. We will try to put more emphasis on these points in the concerned paragraphs.

---

### Official Review · Reviewer_cf8v · 2024-03-22

**Q2-1 Originality-Novelty:** 3
**Q2-2 Correctness-Technical Quality:** 3
**Q2-5 Clarity Of Writing:** 3

**Q1 Summary And Contributions:**

The paper introduces the notion of algebraic mass function (and of algebraic Moebius transform) and provides a new algebraic
expression for expected utility based on such functions. It shows that this is a generalisation of many other decision rules models.

I acknowledge to have read the authors rebuttal.

**Q2-3 Extent To Which Claims Are Supported By Evidence:**

3: Good: the main claims are supported by convincing evidence (in the form of adequate experimental evaluation, proofs, (pseudo-)code, references, assumptions).

**Q2-4 Reproducibility:**

3: Good: key resources (e.g. proofs, code, data) are available and key details (e.g. proofs, experimental setup) are sufficiently well-described for competent researchers to confidently reproduce the main results.

**Q3 Main Strengths:**

- a nice  generalisation of many other decision rules models

**Q4 Main Weakness:**

none

**Q5 Detailed Comments To The Authors:**

The paper provides a convincing  generalisation of many other decision rules models. However, there are some issues to be addressed here and there

- I believe that \Omega should be finite since the very beginning and overal. In general, be carefull about the finiteness/non-finitenesses of sets. So, also MS(U) should be set of finite multisets, otherwise e.g., in prop.4 mean(X), for X non finite is ill defined.
- p.7. in E^Pess, E^Pess', what are the first three paprameters exactly?

**Q9 Complying With Reviewing Instructions:**

Yes

---

> ### Author Rebuttal · Authors · 2024-04-03
>
> We thanks a lot the reviewer for the review. As to the two questions:
>
> Yes indeed, Omega and X are supposed to be finite - this is introduced in definition 6, and assumed throughout the paper. We will emphasize this point, including in the introduction. Thanks a lot.
>
> The question of a similar model for the infinite case would, at least from a theoretical standpoint, be one of the directions for future research to explore.
>
> The first three parameters in the expressions of E^{pess} et   E^{pess} ' are the domains of valuations used to evaluate degrees of belief, degrees of utility of consequences, and those of acts - thus, within the framework of qualitative utilities, the scale  \Lambda (that is, in most articles, the scale [0,1]).

---

### Meta-Review · Area_Chair_ehXU · 2024-04-16

The paper presents an algebraic representation of decision functions, that has a significant expressive power as it encompasses many known decision rules from the literature.

The work is theoretical yet quite interesting, with some potential to allow for the development of new decision rules. All reviewers did go for an accept, except for a reviewer that provided a review without much specific arguments (I therefore decided to disregard the score, given the review quality).